# Synthesis, Structural Versatility, Magnetic Properties, and I^−^ Adsorption in a Series of Cobalt(II) Metal–Organic Frameworks with a Charge-Neutral Aliphatic (O,O)-Donor Bridge

**DOI:** 10.3390/nano13202773

**Published:** 2023-10-16

**Authors:** Ksenia D. Abasheeva, Pavel A. Demakov, Evgeniya V. Polyakova, Alexander N. Lavrov, Vladimir P. Fedin, Danil N. Dybtsev

**Affiliations:** 1Nikolaev Institute of Inorganic Chemistry SB RAS, 3 Lavrentiev Ave., Novosibirsk 630090, Russia; k.abasheeva@g.nsu.ru (K.D.A.); e.polyakova.niic@gmail.com (E.V.P.); lavrov@niic.nsc.ru (A.N.L.); cluster@niic.nsc.ru (V.P.F.); 2Department of Natural Sciences, Novosibirsk State University, 2 Pirogova St., Novosibirsk 630090, Russia

**Keywords:** coordination polymers, metal–organic frameworks, cobalt(II), N-oxides, magnetization, ion exchange, iodide capture, intermolecular interactions

## Abstract

Four new metal–organic frameworks based on cobalt(II) salts and 1,4-diazabicyclo[2.2.2]octane N,N’-dioxide (odabco) were obtained. Their crystallographic formulae are [Co_3_(odabco)_2_(OAc)_6_] (**1**, OAc^−^ = acetate), [Co(H_2_O)_2_(HCOO)_2_]·odabco (**2**), [Co_2_(H_2_O)(NO_3_)(odabco)_5_](NO_3_)_3_·3.65H_2_O (**3**), and [Co_2_(DMF)_2_(odabco)_4_](NO_3_)_4_·3H_2_O (**4**; DMF = N,N-dimethylformamide). Crystal structures of **1**–**4** were determined by single-crystal X-ray crystallography. Coordination polymer **1** comprises binuclear and mononuclear metal–acetate blocks alternating within uncharged one-dimensional chains, in which odabco acts as a bridging ligand. A layered Co(II) formate **2** contains odabco only as guest molecules located in the interlayer space. Layered compound **3** and three-dimensional **4** have cationic coordination frameworks with 26% and 34% specific void volumes, respectively, unveiling high structural diversity of Co(II)-odabco MOFs based on quite a rare aliphatic moiety. Magnetization measurements were performed for **1**, **3**, and **4** and the obtained data were interpreted on the basis of their crystal structures. A strong (J/k_B_~100 K) antiferromagnetic coupling was found within binuclear metal blocks in **1**. Ion exchange experiments revealed a considerable iodide uptake by **3** resulting in an up to 75% guest nitrate substitution within the voids of a coordination framework, found by capillary zone electrophoresis data and confirmed by single-crystal XRD. A preservation of **3** crystallinity during the exchange allowed for the guest I^−^ positions within a new adduct with the formula [Co_2_(H_2_O)(NO_3_)(odabco)_5_]I_2_(NO_3_)·1.85H_2_O (**3-I**) to be successfully determined and the odabco aliphatic core to be revealed as a main adsorption center for quite large and easily polarizable iodide anions. In summary, this work presents a comprehensive study for a series of 1,4-diazabicyclo[2.2.2]octane N,N’-dioxide-based MOFs of cobalt(II) within the framework of magnetic properties and reports the first example of anion exchange in odabco-based coordination networks, supported by direct X-ray structural data. The reported results unveil promising applications of such frameworks bearing ligands with an aliphatic core in the diverse structural design of selective adsorbents and other types of functional materials.

## 1. Introduction

Metal–organic frameworks (MOFs) represent a rapidly developing class of hybrid materials consisting of clusters or metal ions connected by organic bridging ligands. The compounds of this class possess regular polymeric structures, high porosities, and large surface areas. Increasing attention is drawn to the broad prospects for the use of MOFs in catalysis [1,2], selective adsorption [3,4,5], gas separation [6,7,8,9], sensors [10,11,12], and magnetic materials [13]. MOFs with a charged framework necessarily include counter ions in their structure, which may be substituted for other charged species, thus providing numerous applications in ion exchange or selective ion capture. Increasing attention is drawn to natural pollutants existing in anionic forms, such as toxic oxoanions of *d*-metals [11,14,15], *p*-elements [16,17,18], and several dyes [19]. Among other dangerous species, radioactive iodide I^−^ anions provide one of the greatest threats since iodine is an indispensable biological microelement with high environmental mobility. Iodine has several radioactive isotopes, which are largely formed as nuclear reactor waste, including short-lived ^131^I with a half-life time of τ_1/2_ ≈ 8 days and long-lived ^129^I (τ_1/2_ ≈ 16 million years). Accumulating of radioactive iodine isotopes in human organs, e.g., the thyroid gland, results in dangerous diseases such as endocrine system disorder and cancer. On the other hand, radioactive iodine isotopes have a variety of medical applications, including imaging of the thyroid and cancer and other organ disease treatment. In such a context, selective adsorption and accumulation of iodine anions directly relate to a number of very important challenges in environmental and human health protection. Substantial progress in the adsorption and detection of I_2_ iodine molecules by porous MOFs and related materials has been achieved recently [20,21,22,23], but the adsorption of anionic forms of iodine by MOFs is not as developed [24,25,26]. Most unfortunately, the available literature usually reports analytical data only, such as the iodine adsorption capacity, but lacks fundamental structural information of iodine species inside the pores of MOFs that not only provides valuable insight on the nature of the adsorption sites or specific host–guest interactions but also rationalizes the design of more efficient iodine/iodate adsorbents.

The typical design of positively charged porous MOFs involves the connection of metal cations by charge-neutral organic molecule linkers. The vast majority of such linkers is represented by N-donor heterocycles, such as 4,4′-bipyridyl or dabco [27,28,29], whereas the O-donor neutral carbohydrate molecules [30,31,32], polyethers [33], and amide derivatives [34,35,36] rarely afford stable MOF structures except for the those based on strongly oxophilic metal cations. N-oxide molecules feature a substantial negative charge on oxygen atoms [37,38], which makes them stronger than N-donor ligands and attractive linkers for the design of chemically stable and robust cationic frameworks based on metal cations of different natures [39,40,41,42]. Despite the obvious advantages of the N-oxide linkers, successful reports of the synthesis of the corresponding MOFs are uncommon, as is the investigation of ion exchange or other functional properties (e.g., luminescence [43,44], magnetic properties [45,46]) of such compounds. One of the reasons for this is the higher degree of conformational mobility of such ligands [47,48] in the structure of the MOF, which generally reduces the probability of the formation of the regular crystalline network. In this regard, the preparation of new cationic MOFs based on N-oxide molecules as well as systematic investigation of their stability and functional properties have a significant impact on the development of such an important class of materials. In the current work, a preparation and structural characterization of four new MOFs based on Co(II) cations and dabco-N,N’-dioxide linkers is reported. The magnetic properties of the products were carefully studied in a broad temperature range, revealing a strong antiferromagnetic coupling, also rationalized in terms of the crystal structure. More importantly, the first example of I^−^ anion exchange was carried out within a cationic N,N’-dioxide-based coordination framework, showing substantial capacity and decent absorption selectivity towards iodide ions. Direct single-crystal X-ray diffraction data reveal the actual I^−^ binding sites and provide valuable information on the driving factors of the selective ion adsorption.

## 2. Materials and Methods

ATTENTION: Care should be taken in all of the procedures carried out with 1,4-diazabicyclo[2.2.2]octane N,N’-dioxide tris-(hydrogen peroxide) solvate (odabco·3H_2_O_2_) and odabco compounds with potentially explosive anions such as nitrate.

### 2.1. Materials

Ligand precursors odabco·3H_2_O_2_ [49] and odabco·HNO_3_ [50] were synthesized as described previously. Other reagents were commercially available and used as received without further purification. Distilled water was used in all of the synthetic experiments. Double-deionized water was used in the sample preparation for capillary zone electrophoresis (CZE) experiments.

### 2.2. Instruments

Infrared (IR) spectra were obtained in the 4000−400 cm^−1^ range using a Bruker Scimitar FTS 2000 spectrometer (Bruker, Billerica, MA, USA) in KBr pellets. Elemental CHNS analyses were carried out using a VarioMICROcube device (Elementar Analysensysteme GmbH, Hanau, Germany). Powder X-ray diffraction (PXRD) data were acquired on Shimadzu XRD-7000 (Shimadzu, Kyoto, Japan) or Bruker D8 Advance diffractometers (Co-Kα radiation, λ = 1.78897 Å for the as-synthesized **3** and **3-I** samples and Cu-Kα radiation, λ = 1.54178 Å for the rest of compounds) at room temperature. Thermogravimetric analysis was carried out using a Netzsch TG 209 F1 Iris instrument (Netzsch, Waldkraiburg, Germany) under Ar flow (30 cm^3^∙min^−1^) at a 10 K∙min^−1^ heating rate. Capillary zone electrophoresis was performed using a Capel 103R device (Lumex Instruments, Saint Petersburg, Russia) equipped with a photometric detector at a wavelength of 254 nm and chromate electrolytes. The solid samples were digested in double-deionized water before the measurements.

Magnetization measurements were carried out using a Quantum Design MPMS-XL SQUID magnetometer (Quantum Design, San Diego, CA, USA) in the temperature range of 1.77–330 K in magnetic fields up to 10 kOe. To test the thermomagnetic reversibility, temperature dependences of the magnetization M(T) were measured upon heating the sample after it had been cooled either in a zero magnetic field or in a given magnetic field as well as upon cooling the sample. In order to determine the paramagnetic component of the molar magnetic susceptibility χ_p_(T), the temperature-independent diamagnetic contribution χ_d_ and a possible magnetization of ferromagnetic micro-impurities χ_FM_(T) were evaluated and subtracted from the measured values of the total molar susceptibility χ = M/H. Whereas χ_d_ was calculated using Pascal’s additive scheme, χ_FM_(T), if any, was determined from the measured isothermal M(H) dependencies and the M(T) data taken at different magnetic fields. To determine the effective magnetic moment of cobalt Co(II) ions µ_eff_, the paramagnetic susceptibility χ_p_(T) was analyzed using the Curie–Weiss dependence $\chi_{p}\left(T\right)=N_{A}\mu_{\mathrm{eff}}^{2}/3k_{B}\left(T-\theta \right)$, where N_A_ and k_B_ are the Avogadro and Boltzmann numbers, respectively.

Diffraction data for the single crystals of **1** and **3-I** were collected on the “Belok” beamline [51,52] (λ = 0.74503 Å) of the National Research Center Kurchatov Institute (Moscow, Russian Federation) using a Rayonix SX165 CCD detector (Rayonix, Evanston, IL, USA). The XDS program package [53] was used for the data indexing, integration, scaling, and absorption correction. Diffraction data for the single crystals of **2**–**4** were collected on an automated Agilent Xcalibur diffractometer equipped with an AtlasS2 area detector and a graphite monochromator (λ(MoKα) = 0.71073 Å). The CrysAlisPro 1.171.40.84a program package [54] was used for indexing, integration, scaling, and absorption correction. Structures were solved by a dual-space algorithm in SHELXT [55] and refined in an anisotropic approximation (except H atoms) with a full-matrix least squares method in SHELXL [56]. The positions of the hydrogen atoms in the organic ligands were calculated geometrically and refined in the riding model. The PLATON SQUEEZE [57] procedure was implemented for the subtraction of the non-ordered solvent structure factors’ contribution in **4** and **3-I**. Details of the single-crystal structure determination experiments and structure refinements are summarized in the ESI (Appendix A and Page 3). CCDC 2251753–2251757 entries contain the supplementary crystallographic data for this paper. These data can be obtained free of charge from The Cambridge Crystallographic Data Center at https://www.ccdc.cam.ac.uk/structures/, accessed on 13 October 2023. Hirshfeld surfaces and intermolecular contact fingerprints were built in CrystalExplorer 2.1 software [58].

### 2.3. Synthetic Methods

Synthesis of [Co_3_(odabco)_2_(OAc)_6_] (**1**). Cobalt(II) acetate tetrahydrate (24.9 mg, 0.10 mmol) and odabco·3H_2_O_2_ (16.2 mg, 0.067 mmol) were mixed in a glass vial, dispersed in 2.5 mL of DMF, and acidified by 0.11 mL (1.9 mmol) of acetic acid. The mixture in a screw-capped vial was treated in an ultrasonic bath until the reagents were completely dissolved in 15 min and then kept at 80 °C for 10 days. Yield: 57 mg (52%). IR, ν/cm^−1^: 3400 (w), 3030 (s), 1565 (w), 1419 (w), 1347 (s), 1183 (s), 1096 (m), 1047 (m), 1020 (m), 896 (m), 852 (m), 676 (w), 617 (m), 525 (s), 483 (s), 460 (s), 420 (s). Elemental analysis results found (%): C, 35.2; H, 5.2; N, 7.0. Calculated for [Co_3_(odabco)_2_(OAc)_6_] (C_24_H_42_Co_3_N_4_O_16_): C, 35.2; H, 5.2; N, 6.8. TGA: 38.5% weight loss at 230 °C (calculated for 2odabco—35%)

Synthesis of [Co(H_2_O)_2_(HCOO)_2_]·odabco (**2**). Cobalt(II) nitrate hexahydrate (29.1 mg, 0.10 mmol) and odabco·HNO_3_ (62.1 mg, 0.30 mmol) were mixed in a glass vial, dispersed in a mixture of DMF (5.0 mL) and water (0.40 mL), and then acidified with 25 µL (0.42 mmol) of nitric acid 62% water solution. The mixture was treated in a screw-capped vial in an ultrasonic bath until the reagents were completely dissolved in 15 min and kept at 100 °C for 20 h. Yield: 32 mg (97%). IR, ν/cm^−1^: 3046 (s), 2835 (s), 2692 (s), 2497 (s), 2161 (s), 1946 (s), 1577 (w), 1475 (m), 1458 (s), 1347 (w), 1100 (w), 1053 (m), 1022 (s), 895 (s), 856 (m), 810 (s), 790 (w), 677 (w), 483 (m), 453 (w). Elemental analysis results found (%): C, 29.1; H, 5.5; N, 8.5. Calculated for [Co(H_2_O)_2_(HCOO)_2_]·odabco (C_8_H_18_CoN_2_O_8_): C, 29.1; H, 5.5; N, 8.5. TGA: 12% weight loss at 150 °C (calculated for 2H_2_O—11%).

Synthesis of [Co_2_(H_2_O)(NO_3_)(odabco)_5_](NO_3_)_3_·3.65H_2_O (**3**). Cobalt(II) nitrate hexahydrate (29.1 mg, 0.10 mmol) and odabco·3H_2_O_2_ (74.0 mg, 0.30 mmol) were mixed in a glass vial, dispersed in a mixture of DMF (5.0 mL) and water (0.40 mL) and then acidified with 25 µL (0.42 mmol) of nitric acid 62% water solution. The mixture was treated in a screw-capped vial in an ultrasonic bath until the reagents were completely dissolved in 15 min and kept at 70 °C for 72 h. Yield: 36.5 mg (63%). IR, ν/cm^−1^: 3409 (w), 3255 (s), 3032 (m), 2917 (s), 2485 (s), 2404 (s), 2118 (s), 1923 (s), 1648 (m), 1465 (w), 1353 (w), 1180 (s), 1092 (w), 1040 (m), 963 (s), 895 (w), 852 (m), 824 (s), 806 (s), 677 (w), 606 (s), 531 (s), 483 (s), 459 (s). Elemental analysis results found (%): C, 31.1; H, 6.1; N, 16.3. Calculated for [Co_2_(H_2_O)(NO_3_)(odabco)_5_](NO_3_)_3_·4.5H_2_O·0.5DMF (C_31.5_H_74.5_Co_2_N_14.5_O_28_): C, 31.0; H, 6.1; N, 16.6. TGA: 10% weight loss at 115 °C (calculated for 5.5H_2_O + 0.5DMF—11%); explodes at 250 °C.

Synthesis of [Co_2_(DMF)_2_(odabco)_4_](NO_3_)_4_·3H_2_O (**4**). Cobalt(II) nitrate hexahydrate (29.1 mg, 0.10 mmol) and odabco·3H_2_O_2_ (74.0 mg, 0.30 mmol) were mixed in a glass vial, dispersed in a mixture of DMF (5.0 mL) and water (0.40 mL), and then acidified with 25 µL (0.42 mmol) of nitric acid 62% water solution. The mixture was treated in a screw-capped vial in an ultrasonic bath until the reagents were completely dissolved in 15 min and kept at 60 °C for 20 h. Yield: 50 mg (90%). IR, ν/cm^−1^: 3440 (w), 3037 (m), 2944 (s), 2886 (m), 2806 (s), 2497 (s), 2390 (s), 2350 (s), 2292 (s), 2120 (s), 1920 (s), 1750 (s), 1654 (w), 1476 (m), 1442 (s), 1394 (m), 1351 (w), 1255 (s), 1181 (s), 1090 (w), 1045 (m), 966 (s), 897 (w), 854 (m), 825 (m), 682 (w), 552 (m), 498 (m), 455 (s). Elemental analysis results found (%): C, 31.5; H, 6.1; N, 16.8. Calculated for [Co_2_(DMF)_2_(odabco)_4_](NO_3_)_4_]·3.5H_2_O (C_30_H_69_Co_2_N_14_O_25.5_): C, 31.3; H, 6.0; N, 17.0 TGA: 5% weight loss at 100 °C (calculated for 3.5H_2_O—5%); 13.5% weight loss at 155 °C (calculated for 2DMF—13%); explodes at 250 °C.

Synthesis of [Co_2_(H_2_O)(NO_3_)(odabco)_5_]I_2_(NO_3_)·1.85H_2_O (**3-I**). A ca. 50 mg portion of fresh **3** crystals was immersed in 25.0 mL of 0.1M sodium iodide (NaI) solution in a 95:5 (*v*/*v*) mixture of DMF and H_2_O in a screw-capped vial without access to sunlight. Yield is close to quantitative. IR, ν/cm^−1^: 3450 (w), 3034 (s), 3010 (w), 2954 (s), 2905 (s), 2781 (s), 2501 (s), 2450 (s), 2357 (s), 2317 (s), 2163 (s), 2108 (s), 1914 (s), 1798 (s), 1641 (m), 1459 (m), 1387 (w), 1180 (s), 1109 (m), 1094 (w), 1040 (m), 963 (s), 894 (w), 858 (w), 827 (m), 683 (w), 509 (m), 465 (m). Capillary zone electrophoresis (CZE) results: I^−^:NO_3_^−^ ratio found: 1.3:1.0. Calculated for 75% guest nitrate substitution ([Co_2_(H_2_O)(NO_3_)(odabco)_5_]I_2.25_(NO_3_)_0.75_·xH_2_O) 1.3:1.0. TGA: 15% weight loss at 115 °C (calculated for 12H_2_O—15%); explodes at 250 °C; 10% weight residue at 400 °C (calculated for 2CoO—10%).

Synthesis of iodide-exchanged samples. Ca. 50 mg portions of fresh **3** crystals were immersed in 25.0 mL of NaI solutions in 95:5 (*v*/*v*) mixtures of DMF and H_2_O in screw-capped vials without access to sunlight with varying NaI concentrations and keeping times, listed in Appendix A. After the immersion and subsequent filtration, the resulting NO_3_^−^:I^−^ ratios in solid samples were determined by CZE and then recalculated to the substitution degrees of the guest nitrate by iodide.

## 3. Results and Discussion

### 3.1. Synthesis and Crystal Structures

Although hydrogen peroxide, presenting in the easily accessible odabco solvate [49,59] used in the syntheses, is a potentially strong oxidizer, no Co^2+^ oxidation was observed in any of the obtained compounds, based on their colors (Appendix A), Co–O bond lengths (Appendix A), and magnetization data (see below). The stability of the cobalt(II) oxidation state is attributed to having used acidic modulators (acetic or nitric acids) in the optimized synthetic methods, providing very high Co^3+^/Co^2+^ reduction potential. Typically, a rapid formation of brown or black precipitates, which were contained Co(III) hydrated oxide phases, was observed in the related systems with neutral or basic media.

Compound [Co_3_(odabco)_2_(OAc)_6_] (**1**) was synthesized from a mixture of cobalt(II) acetate, odabco·3H_2_O_2_, and acetic acid in DMF, which was heated at 80 °C for 10 days. The powder X-ray diffraction (PXRD) pattern of the filtered sample **1** (Appendix A) matched the theory well, indicating a phase purity of the compound. According to the single-crystal X-ray diffraction (SCXRD) data, **1** crystallized in a monoclinic crystal system with a *P*2_1_/*c* space group. The asymmetric unit contained three independent metal ions. Both Co(1) and Co(2) had a square pyramidal oxygen environment comprised of four O atoms from four (μ-κ^1^,κ^1^)-acetates in the pyramid base and one O atom of (μ-κ^1^,κ^1^)-odabco as a top vertex. A pair of symmetry-equivalent metal ions (Co(1)-Co(1) or Co(2)-Co(2)) formed typical “paddlewheel” units (Figure 1a). Co(3) adopted a distorted tetrahedral environment consisting of two O atoms of non-bridging (κ^1^)-acetates and two O atoms of odabco bridges. A fifth close Co…O contact existed between Co(3) and the second O atom of the acetate, providing a trigonal bipyramidal distortion in the geometry of the metal ion (Figure 1b). Mononuclear Co^2+^ ions and binuclear paddlewheels alternated within one-dimensional zigzag-like chains with the summary formula {-Co_3_(odabco)_2_(OAc)_6_-}_n_ (Figure 1c). These chains were uncharged and densely packed within the **1** crystal structure, leaving no solvent-accessible volume. The shortest interchain Co···Co distance of 6.66 Å showed a considerable spatial separation of metal blocks within a three-dimensional crystal packing.

Heating a mixture of cobalt(II) nitrate and odabco·HNO_3_ in an acidified mixture of DMF and water at 100 °C for 20 h led to the formation of [Co(H_2_O)_2_(HCOO)_2_]·odabco (**2**). The PXRD pattern of the filtered sample (Appendix A) confirmed its phase purity. According to the SCXRD, **2** crystallized in an orthorhombic crystal system with a *Pnma* space group. Its asymmetric unit contained one independent metal atom. The Co^2+^ ion had an octahedral environment consisting of four O atoms of four (μ-κ^1^,κ^1^)-formates laying along the *ac* plane and two terminal water molecules coordinated almost parallel to the *b* axis. Thus, two-dimensional rectangular layers with a {-Co(H_2_O)_2_(HCOO)_2_-}_n_ formula were formed (Figure 2). The minimal intralayer Co···Co distance was 6.17 Å. The odabco molecule in **2** was not coordinated by metal ion and was located in the interlayer space, bound by hydrogen bonds to the uncharged 2D coordination network. The O(odabco)…H(H_2_O) hydrogen bond lengths were 1.81 and 1.86 Å, showing the good fit of the odabco molecule to the layer geometry. The interplanar distance between the closest layers in the resulting compound with the summary formula [Co(H_2_O)_2_(HCOO)_2_]·odabco was ca. 7.57 Å.

Interestingly, doubling the concentration of the reagents, compared to the synthesis of **2**, while maintaining the solvent composition led to the crystallization of a new metal–organic framework with the crystallographic formula [Co_2_(H_2_O)(NO_3_)(odabco)_5_](NO_3_)_3_·3.65H_2_O (**3**) in a phase-pure form, confirmed by PXRD (Appendix A). This example showed a remarkable effect of the reactant concentrations on the structure and composition of odabco-based MOFs, possibly related to the comparable donor strengths of odabco and carboxylate anions. According to the single-crystal data, **3** crystallized in a monoclinic crystal system with a *P*2_1_/*n* space group. The asymmetric unit contained two independent metal atoms. Both independent Co^2+^ ions adopted a trigonal bipyramidal oxygen environment, including three O atoms from three (μ-κ^1^,κ^1^)-odabco bridges at the pyramid base and one O atom from terminal odabco at one pyramid vertex. The opposite vertex was occupied by an O atom of terminal nitrate in the coordination environment of Co(1), whereas terminal water oxygen coordinated Co(2) in a similar position. Such coordination geometry is relatively rare for Co^2+^ MOF structures compared to tetrahedron, octahedron, and square pyramid geometries [60,61,62,63,64,65]. Each metal center in **3** acted as a three-connected node. The nodes were interlinked into two-dimensional coordination layers with a strongly distorted **hcb** topology (Appendix A). The layers were packed in an ABAB manner (Appendix A), and interlayer hydrogen bonds between terminal odabco ligands and coordinated water molecules occurred. Such hydrogen bonding apparently stabilized a porous crystal structure of **3** by providing the interconnection for the intralayer apertures to form 1D rectangular channels (Figure 3c) with their laterals being paved odabco ligands and additionally by coordinated nitrates. The size of the channels was 4 × 6 Å^2^, and the void volume in the coordination framework of **3** amounted to 1250 Å^3^ per unit cell (26% of the total unit cell volume). Voids were occupied by nitrate anions and guest water molecules. In the 3D crystal packing of **3**, intra- and interlayer distances between the closest Co(II) ions were quite close to each other, and all of these values exceeded 8.33 Å, showing a considerable spatial separation of paramagnetic metal centers.

By lowering the temperature from 70 °C to 60 °C and reducing the heating time, a new crystalline product [Co_2_(DMF)_2_(odabco)_4_](NO_3_)_4_·3H_2_O (**4**) was isolated. The PXRD pattern of the filtered sample (Appendix A) confirmed its phase purity. According to the SCXRD, **4** crystallized in a monoclinic crystal system with a *C*2/*c* space group. The asymmetric unit contained two independent metal atoms. Co(1) adopted a tetrahedral oxygen environment consisting of four O atoms belonging to four (μ-κ^1^,κ^1^)-odabco bridges (Figure 4a). Octahedral Co(2) was additionally coordinated by two O atoms of terminal DMF ligands and thus represented a square node (Figure 4b). Tetrahedral and square nodes were interlinked in a chess-like manner to form a three-dimensional coordination framework (Figure 4c) possessing a well-known **pts** topology (Figure 4d). The voids in the coordination framework of **4** were represented by narrow channels running along the *b* axis with a size of 2 × 2 Å, and its specific void volume amounted to 34%. The voids were occupied by nitrate counter anions and guest water.

### 3.2. Magnetic Properties

Temperature and magnetic-field dependences of the magnetic susceptibility χ of the odabco-based coordination networks **1**, **3**, and **4** were measured in order to shed light on the spin states of paramagnetic metal centers and their magnetic interactions. All of the studied compounds exhibited paramagnetic behavior in the entire temperature range of 1.77–300 K, with χ smoothly growing with decreasing temperature (χ(T) data are provided in Appendix A). The measured χ(T) data were corrected for the temperature-independent diamagnetic contribution to determine the paramagnetic component χ_p_(T) associated solely with the Co(II) ions.

In the case of **3** and **4**, the χ_p_^−1^(T) curves possessed a fairly simple shape typical of compounds with magnetically diluted Co(II) ions and showed no sign of any spin transition (Figure 5a,b). Indeed, in the low-temperature region χ_p_^−1^(T) went straight to the origin, tending to intersect the temperature axis at a temperature θ lower than the experimentally resolved one, |θ| < 0.1 K (solid blue lines in Figure 5a,b). One can therefore infer that the exchange interaction between Co(II) ions was virtually absent and employed the Curie–Weiss description with θ ≈ 0 and a temperature-dependent effective magnetic moment μ_eff_(T) (Figure 5a,b). As can be seen in the figures, μ_eff_ changed from 4.51 μ_B_ and 4.68 μ_B_ at T = 300 K to 3.53 μ_B_ and 3.70 μ_B_ at T = 1.77 K for **3** and **4**, respectively (at high magnetic fields and low temperatures, μ_eff_ drops to lower values due to the non-linear magnetization M(H) dependence). The high-temperature μ_eff_ values observed are typical for Co(II) ions and considerably exceeded the spin-only moment μ_eff_ ≈ 3.87 μ_B_ (for S = 3/2) owing to the rather large contribution from orbital moments. In turn, the decrease in the effective magnetic moment with cooling was apparently due to the zero-field splitting (ZFS) of the ground-state multiplets of the Co(II) ions [66]. Given the presence of independent Co(II) ions with different coordination in the structures of both **3** and **4**, one can hardly determine all of their ZFS parameters by fitting the magnetic data. However, even the simplest model [66], assuming a rather large average axial ZFS parameter D/k_B_ ≈ 50 K and taking the rhombic ZFS parameter E/k_B_ = 0, yielded a reasonably good fit to the μ_eff_(T) data (dashed orange line in Figure 5a). To substantiate the explanation based on ZFS, one can take a look at the field dependences of the magnetization M(H) of **3** and **4** at T = 1.77 K (Appendix A), which clearly demonstrated the single-ion anisotropy generated by ZFS. Indeed, the normalized susceptibility χ(H)/χ(0), whose behavior is governed by the most magnetically productive component χ_||_ or χ_┴_ [66], can be described well (Appendix A) by the conventional expression
$$M\left(H\right)=N_{A}{g\mu }_{B}{\mathrm{SB}}_{S}\left(\frac{{g\mu }_{B}}{k_{B}T}\mathrm{SH}\right)$$(where B_S_(x) is the Brillouin function) for S = 3/2 ions, with the g-factor g = 2.48–2.80 being similar or slightly exceeding the g values evaluated from the high-temperature μ_eff_ data.

In contrast, the absolute values of the magnetization of **3** and **4** turned out to be considerably (by a factor of 1.85–2.15) lower than what would follow from the aforementioned expression (Appendix A), indicating that the magnitude of the least magnetically productive component, χ_||_ or χ_┴_, was drastically suppressed. We can conclude, therefore, that it was ZFS and single-ion anisotropy that stayed behind the observed temperature dependence of μ_eff_ and the low-temperature magnetic properties of compounds **3** and **4**.

The absence of any measurable interactions between Co(II) ions in **3** and **4** agrees well with their crystal structures, which comprised only mononuclear metal blocks. We can conclude, therefore, that odabco aliphatic bridges are incapable of providing effective routes for exchange interactions between Co(II) ions because of their rather long molecular length and the absence of a conjugated π-system.

A more complicated magnetic behavior was found for compound **1**. As can be seen in Figure 5c, the inversed magnetic susceptibility curve χ_p_^−1^(T) demonstrates that there were two quasi-linear sections (at T < 40 K and T > 90 K) separated by a gradual crossover, which was accompanied by a clear step in the μ_eff_(T) dependence. In the low-T region, the effective magnetic moment of **1** calculated per formula unit containing three Co(II) ions (Figure 5c) approached the μ_eff_ values per Co(II) ion in **3** and **4** (Figure 5a,b), which implies that only one out of three Co(II) ions in **1** should have been contributing to the low-temperature magnetization. Even more clear evidence comes from the M(H) data measured for **1** at the lowest temperature of 1.77 K (Appendix A). Both the M(H) and χ(H)/χ(0) curves of **1** were almost indistinguishable from those of **3** (Appendix A). Hence, in the low-temperature limit, one of the three Co(II) ions in **1** behaved exactly like the non-interacting Co(II) ions in sample **3**, whereas the other two Co(II) ions provided no contribution to the magnetization, providing evidence for the singlet state of the pair. Such behavior of **1** can be perfectly explained by its crystal structure, comprising binuclear paddlewheel blocks that contain closely located metal ions and mononuclear Co(II) ions in a 1:1 ratio. The crossover in the χ_p_^−1^(T) and μ_eff_(T) curves observed in the temperature range of 40–90 K demonstrates the transformation of AF-coupled Co(II) dimers from a low-T non-magnetic singlet state into a high-T paramagnetic one.

One can see in Figure 5c that μ_eff_ did not reach its saturation value even at T = 300 K, indicating the presence of a substantial AF exchange interaction within binuclear carboxylate blocks even at room temperature. Given the presence of three independent Co(II) ions with different structural environments, a precise determination of all numerous ZFS and exchange parameters in compound **1** is an extremely difficult task that can by no means be achieved just by fitting the magnetic data and is therefore beyond the scope of this work. However, it turns out that the χ_p_(T) and μ_eff_(T) data could still be described reasonably well within a simplified model, assuming the magnetic susceptibility of **1** to contain two independent contributions: from isolated Co(II) ions and from pairs of Co(II) ions, which belonged to binuclear paddlewheel blocks and were coupled into Co(II)-Co(II) dimers by the AF exchange interaction J [66,67], χ_p_ = χ_iso_ + χ_dim_. The first term, χ_iso_, could be evaluated by analyzing the data in the low-T region, where Co(II) dimers occupied the singlet ground state and provided a negligible contribution to the magnetization. Following this approach, we fitted the model of isolated Co(II) ions with ZFS to the low-T χ_p_ data and estimated the axial ZFS parameter D/k_B_ of these ions to be ≈ 20 K; the corresponding contribution of isolated Co(II) ions to μ_eff_(T) is shown by the dashed orange line in Figure 5c. In its turn, the contribution of AF-coupled Co(II) dimers χ_dim_ could be roughly approximated by the formulae established in Ref. [66] for dimers with ZFS in the strong exchange limit (|J| >> |D|). The best-fitting result was obtained for the AF exchange interaction J/k_B_~100 ± 20 K, with the uncertainty being related to the selection of D values. As can be seen in Figure 5c, a sum of the fitted contributions from isolated Co(II) ions and Co(II) dimers (dashed cyan line) yielded a fair account of the μ_eff_(T) behavior of **1**.

It is worth emphasizing that in compound **1** the Weiss constant determined in the low-temperature region was also negligibly small, |θ| < 0.1 K, indicating no sizable exchange interaction between either the paddlewheels and residual isolated metal ions or between the polymeric chains. From the point of view of magnetic properties, therefore, compound **1** can be considered a collection of paramagnetic Co(II) ions and AF-coupled Co(II)-Co(II) dimers in a 1:1 ratio that do not interact with each other.

### 3.3. Anion Exchange

According to the X-ray structural data, the cationic coordination framework [Co_2_(H_2_O)(NO_3_)(odabco)_5_]^3+^ in **3** featured sufficient void space of up to 26% (*v.*/*v.*) and a channel size of ca. 4 × 6 Å for a facile diffusion of single-atom anions such as iodides (ionic diameter ≈ 4.2 Å), which has great practical relevance. Due to a charge balance, the adsorption of the iodide ions needed to be accompanied by the substitution of the uncoordinated NO_3_^−^ anions (wide infra). The ion exchange was tested by immersion of the crystals of **3** in different solutions of NaI in a DMF/H_2_O mixture with varying concentrations and exposure times followed by a simultaneous analysis of the iodide and nitrate ions via capillary zone electrophoresis (CZE, Appendix A). A remarkable degree of NO_3_^−^ substitution equal to 75% occurred in 0.1M solution of NaI after one week. This ion exchange ratio corresponds to the chemical formula [Co_2_(H_2_O)(NO_3_)(odabco)_5_]I_2.3_(NO_3_)_0.7_ or 27 wt.% of the iodine gravimetric uptake. This number is rather close to the upper theoretical limit of 33 wt.% of the iodine capacity, assuming complete ion exchange. At lower NaI concentrations and/or shorter exposure time, the iodine uptake decreased as expected, with ca. 1% of the nitrate ion substitution achieved after one day of adsorption from the diluted 10^−4^ M NaI solution (Appendix A). The reversibility of I^−^ adsorption was also confirmed by a consecutive immersion of **3** in 0.1M NaI and then in 0.1 M NaNO_3_ solutions with a subsequent CZE analysis of the samples (Appendix A), which showed an iodide exchange back to the nitrate ions of at least 89%.

Although the quality of the single crystals of **3** decreased during the anion exchange experiments, we managed to perform an SCXRD of the iodide-saturated sample **3-I**, which allowed for a detailed analysis of the positions of the incorporated I^−^ anions. According to the crystal structure of **3-I**, no coordination of the iodide to Co^2+^ was observed, as the coordinated NO_3_^−^ anions resided almost in the same place. Inside the pores of **3-I**, six different positions of the guest I^−^ were directly localized, each with 50% occupancy, corresponding to 1.5 symmetrically independent iodide ions per formula unit. An assignment of the other electron density peaks, as well as the analysis of the SQUEEZE data (see page 3 in the ESI), gave rise to the meaningful chemical formula [Co_2_(H_2_O)(NO_3_)(odabco)_5_]I_2_(NO_3_)·1.85H_2_O, which is also concordant with the chemical analyses and very close to the above composition. For all of the located iodide ions, multiple C–H···I contacts were found (Figure 6 and Appendix A), suggesting that the aliphatic core of the odabco ligands furnished a particularly favorable environment for the iodide anions. Also, a relatively short O···I distance of 3.46 Å was identified between one of the I^−^ anions (I2) and guest H_2_O molecules (O2W), implying a possible H-bond interaction as an additional factor stabilizing the position of the iodide (Figure 6b). The Hirshfeld surface analysis (Figure 6c), as well as intermolecular contact fingerprints (Appendix A), provided more detailed information on the nature of the interatomic contacts. According to Table 1, the I···H contacts largely dominated among all of the notable interatomic interactions, contributing to at least 94% of the Hirshfeld surface of every independent iodide position, quantitatively confirming the decisive role of the aliphatic ligand in the stabilization of the I^−^ adsorption centers. In other words, the substitution of the nitrates by the iodide ions in **3** was mainly driven by seemingly weak but numerous van der Waals interactions between I^−^ and the H atoms of the odabco ligands, resulting from an optimal geometrical fit of the large and easily polarizable spherical iodide anion in the interstitial positions of the mostly aliphatic coordination framework.

To ensure that the ion adsorption was selective, various anions were tested for the exchange reaction, and the results fell into two categories depending on the size and the coordination ability of the anion. The crystals of **3** were seemingly stable and showed no detectable presence of the other ionic species while immersed in the solutions of the larger multinuclear ions I_3_^−^, NO_2_^−^, CN^−^, SeO_4_^2−^, CrO_4_^2−^, and Cr_2_O_7_^2−^. On the contrary, a deterioration of the compound was observed in the solutions of mononuclear halide anions F^−^, Cl^−^, and Br^−^, as well as azide N_3_^−^. Supposedly, the chemical and structural transformations of **3** in ionic solutions were first determined by the size of the anion. The multinuclear species, whose size exceeded the aperture of the channels of the coordination framework, could not participate in the anion exchange and enter the structure. The mononuclear halide anions obviously had no steric limitations and could penetrate the channels through the substitution of the uncoordinated nitrate ions. The framework degradation was likely invoked by the interaction of those halide or azide anions with Co(II) cations, plausibly through the substitution of the coordinated nitrate ligands, as the metal centers in **3** had no other coordination sites available. The iodide seems to have occupied quite a unique position among the anions under study. Its relatively small size admitted facile adsorption through the exchange of the interstitial NO_3_^−^ ions, whereas its poor electron donor ability prevented the substitution of NO_3_^−^ ligands in the coordination sphere of the Co(II) cations; therefore, framework **3** remained intact during the ion exchange processes, as unambiguously confirmed by powder XRD for **3-I** (Appendix A). Moreover, the long-term stability of the crystalline material even increased after the iodide adsorption since the powder XRD plot of **3-I** was reproduced after storage of the sample for six months (Figure 7), whereas the original compound **3** underwent a phase transition under the same conditions. Apparently, an odabco-lined pore environment of the framework created a favorable confinement for the large polarizable I^−^ ion through multiple interatomic interactions, which not only drove the iodide ion exchange/adsorption but also improved the stability of both the local I^−^ adsorption sites and a whole crystalline framework. High structural and functional durability of the adsorbent is essential for the safe storage of long-lived iodine radionuclides such as ^125^I, ^129^I, and ^131^I in stable solid forms as well as on-demand release of those species due to the reversibility of iodide-nitrate ion exchange. Overall, the porous cationic material **3** developed in this work offers targeted adsorption of the iodide anions, including the radioactive species, from complex solutions in various concentrations; rather capacious and long-term storage; and convenient extraction of the iodine when necessary.

## 4. Conclusions

In summary, four new metal–organic frameworks based on cobalt(II) and 1,4-diazabicyclo[2.2.2]octane N,N’-dioxide (odabco) were successfully synthesized and structurally characterized. Based on synthetic conditions, the obtained compounds possessed different Co(II) coordination environments, network charges, dimensionality, and porosity, revealing a diverse role of such ligands in MOF chemistry. The temperature-dependent magnetization measurements of two samples demonstrated their paramagnetic nature, resulting from a spatial separation of metal blocks with no sizable exchange interaction between Co(II) ions. However, in [Co_3_(odabco)_2_(OAc)_6_], a strong (J/k_B_~100 K) antiferromagnetic coupling was revealed in the binuclear Co(II)-carboxylate blocks that persisted in the crystal structure along with the mononuclear units. Anion exchange was investigated for a porous and stable cationic MOF [Co_2_(H_2_O)(NO_3_)(odabco)_5_](NO_3_)_3_. A remarkable 27 wt.% iodide uptake was achieved via substitution of guest nitrates by I^−^ anions, accompanied by full preservation of the crystallinity of the sample during the exchange. Iodide positions were directly determined via the single-crystal XRD method within an anion-exchanged adduct [Co_2_(H_2_O)(NO_3_)(odabco)_5_]I_2_(NO_3_)·1.85H_2_O. According to the analysis of structural data, the adsorption sites of the iodide anions were lined by the aliphatic core of the odabco ligand, providing a mostly aliphatic environment. The selective and reversible adsorption of the I^−^ among other anions resulted from a delicate interplay between the size of the iodide anion, which was small enough for facile diffusion along the 3D crystal structure and large enough to enjoy multiple non-covalent interactions with the odabco molecules, and the poor coordination ability of the I^−^ ions. Such insights would not be possible without comprehensive structural characterization of the iodide exchange compound. The obtained data rationalize the experimental observations and provide generalized solutions for the targeted design of more effective ion adsorbent materials. Also, the reported results demonstrate the perspectives of 1,4-diazabicyclo[2.2.2]octane N,N’-dioxide as an aliphatic (O,O)-donor charge-neutral ligand for the structural design of multifunctional metal–organic frameworks, especially for the synthesis of cationic coordination lattices for magnetic applications, anion adsorption, and similar practical tasks.

## Figures and Tables

**Figure 1 nanomaterials-13-02773-f001:**
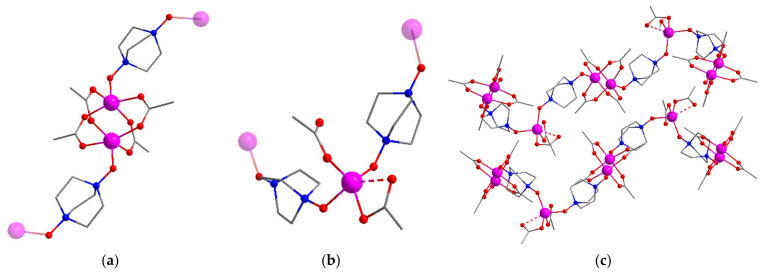
Binuclear paddlewheel (**a**) and mononuclear (**b**) blocks in the structure of **1**. The Co atoms closest to the blocks are shown to be transparent. Two neighboring coordination chains in **1** (**c**). Co atoms are shown in purple; O atoms are red; N atoms are blue. H atoms are not shown.

**Figure 2 nanomaterials-13-02773-f002:**
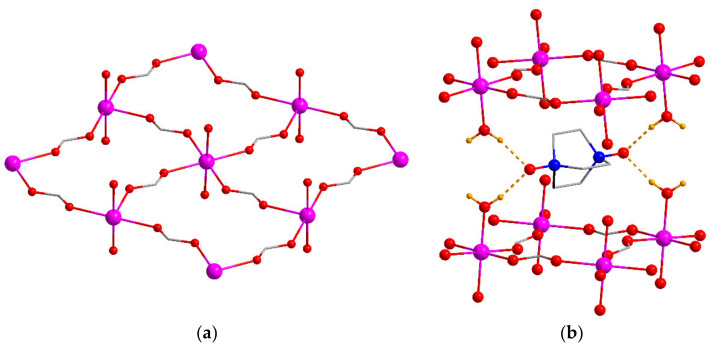
Fragment of the {-Co(H_2_O)_2_(HCOO)_2_-}_n_ coordination layer (**a**) and the cage in the interlayer space with a bound odabco guest (**b**) in **2**. Atom designations match Figure 1, H atoms of water are shown in orange. Hydrogen bonds are shown by orange dashed lines.

**Figure 3 nanomaterials-13-02773-f003:**
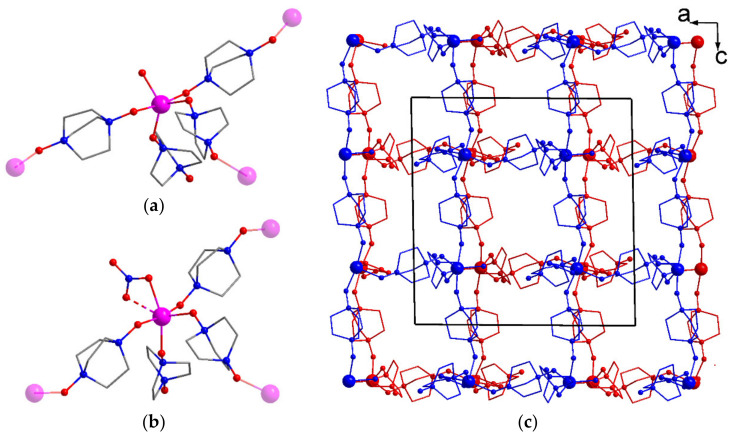
{Co(H_2_O)(O_odabco_)_4_} (**a**) and {Co(NO_3_)(O_odabco_)_4_} (**b**) triconnected nodes in **3**. The Co atoms closest to the nodes are shown as transparent. Atom designations match Figure 1. [Co_2_(odabco)_5_(H_2_O)(NO_3_)]^3+^ coordination network, viewed along the *b* axis (**c**). Two independent layers are shown in different colors. H atoms and guest moieties are not shown. Unit cell edges are shown in black solid lines.

**Figure 4 nanomaterials-13-02773-f004:**
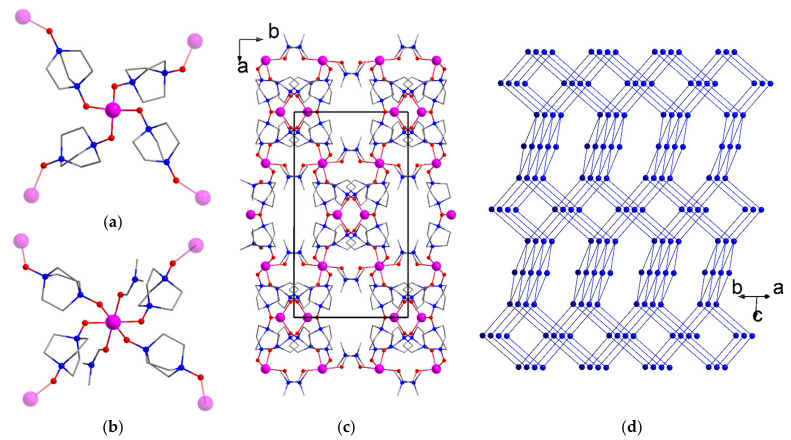
{Co(O_odabco_)_4_} tetrahedral node (**a**) and {Co(DMF)_2_(O_odabco_)_4_} square node in **4** (**b**). Coordination framework of **4**, viewed along the *c* axis (**c**). Atom designations match Figure 1. H atoms and guest moieties are not shown. Unit cell edges are shown in black solid lines. Topological representation of the **4** framework (**d**). Each metal center and each odabco ligand are shown as a node and an edge, respectively.

**Figure 5 nanomaterials-13-02773-f005:**
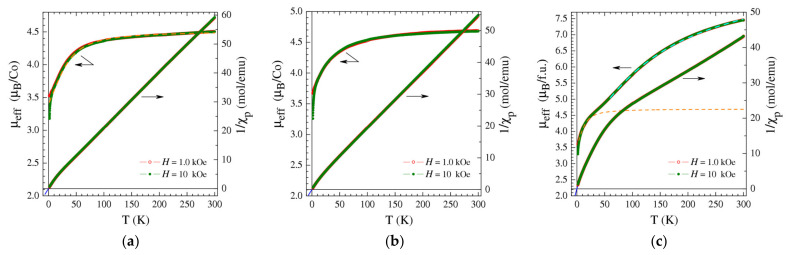
Temperature dependencies of the effective magnetic moment μ_eff_ and inversed magnetic susceptibility 1/χ_p_ measured at magnetic fields H = 1; 10 kOe for **3** (**a**), **4** (**b**), and **1** (**c**). The effective magnetic moment is given per Co(II) ion for **3** (**a**) and **4** (**b**), whereas for **1** (**c**) it is given per formula unit containing 3 Co(II) ions. The solid blue lines in panels (**a**–**c**) show Curie–Weiss fits for the low-temperature 1/χ_p_ data measured at H = 1 kOe. The dashed orange line in panel (**a**) depicts an approximation of the μ_eff_ data, taking into account the ZFS of Co(II) ions with the axial ZFS parameter D/k_B_ ≈ 50 K. In panel (**c**), the dashed orange line shows a ZFS fit for one isolated Co(II) ion; the dashed cyan line shows an approximation of the μ_eff_ data that includes contributions from one isolated Co(II) ion with ZFS and from an AF coupled pair of Co(II) ions with ZFS.

**Figure 6 nanomaterials-13-02773-f006:**
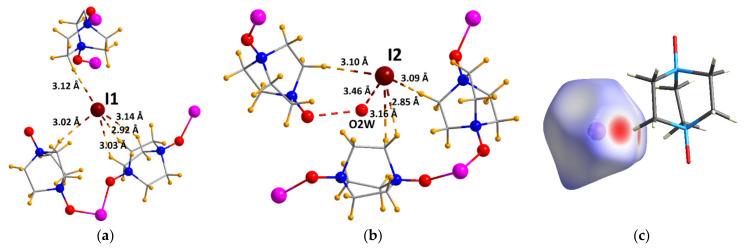
Iodide molecular environment in **3-I** for I1 (**a**) and for I2 (**b**). H atoms are shown in orange. Other atom designations match Figure 1. Hirshfeld surface of I1 (**c**). Only the closest odabco moiety is shown.

**Figure 7 nanomaterials-13-02773-f007:**
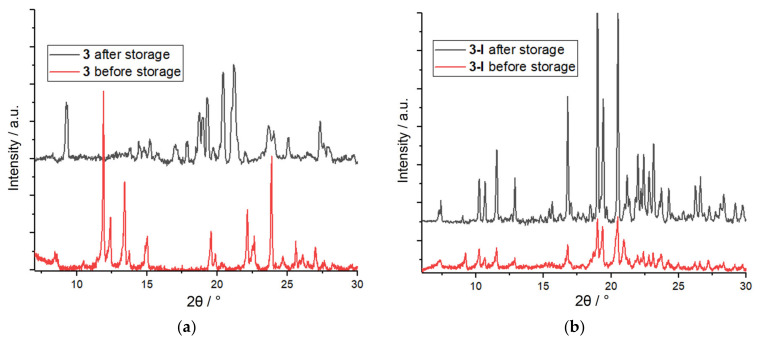
PXRD patterns of **3** (**a**) and **3-I** (**b**) after six months of storage.

**Table 1 nanomaterials-13-02773-t001:** Atom contributions into intermolecular contacts with I atoms.

Atom/Contact	I…H, %	I…O, %	I…N, %	I…I, %	I…Co, %
I1	96.6	3.2	0.2	0.0	0.0
I2	94.8	5.2	0.0	0.0	0.0
I3	98.8	0.6	0.0	0.7	0.0
I4	96.9	3.1	0.0	0.0	0.0
I5	95.0	5.0	0.0	0.0	0.0
I6	94.1	5.3	0.0	0.6	0.0

## Data Availability

CCDC 2251753–2251757 entries contain the supplementary crystallographic data for this paper. These data can be obtained free of charge from The Cambridge Crystallographic Data Center at https://www.ccdc.cam.ac.uk/structures/, accessed on 13 October 2023.

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
