# Peer review of "Synthesis, Structural Versatility, Magnetic Properties, and I Adsorption in a Series of Cobalt(II) Metal–Organic Frameworks with a Charge-Neutral Aliphatic (O,O)-Donor Bridge"

_nanomaterials, 2023, doi:10.3390/nano13202773_

Round 1

Reviewer 1 Report

This manuscript describes the synthesis of cobalt (II) complexes in a metal-organic framework and their potential use as a trapping agent for I- anions.  It is surprising that the peroxide does not oxidise the Co(II) systems but the data is convincing that this does not happen. The results are well presented and the data are well interpreted.  The introduction is good and sets the research in context.  The X-ray and magnetic studies are described in detail.  

My only criticism is the length of X-ray crystallography data collection discussion in the paper (pages 3, 4).  A lot of this information should be placed in the supplementary information.

The English is not perfect but very good and requires very minor editing

Author Response

This manuscript describes the synthesis of cobalt (II) complexes in a metal-organic framework and their potential use as a trapping agent for I- anions. It is surprising that the peroxide does not oxidise the Co(II) systems but the data is convincing that this does not happen. The results are well presented and the data are well interpreted. The introduction is good and sets the research in context. The X-ray and magnetic studies are described in detail.

Authors are grateful to the reviewer for high evaluation of the work.

My only criticism is the length of X-ray crystallography data collection discussion in the paper (pages 3, 4). A lot of this information should be placed in the supplementary information.

The corresponding text has been shortened and partially transferred into ESI (page 3).  

Reviewer 2 Report

The paper is scientifically sound and well presented with results supporting conclusions. 

The material reported is somewhat derivative being an extension of the authors previous work with dabco ligands and other metals. 

The application to iodine capture / exchange is novel however. There are two small suggestions to be made; one which is relatively simple and could be included, the other less so. To be relevant to the absorption of iodide ions it would be interesting to understand the selectivity. If the material is placed in a solution containing a mixture of anions which are absorbed preferentially? Is there selectivity in a complex mixture? 

More challenging would be to study the structural damage / integrity once radioisotopic iodide is absorbed.   

a few minor english issues require attention.

Author Response

The paper is scientifically sound and well presented with results supporting conclusions. The material reported is somewhat derivative being an extension of the authors previous work with dabco ligands and other metals. The application to iodine capture / exchange is novel however. There are two small suggestions to be made; one which is relatively simple and could be included, the other less so.

Authors are grateful to the reviewer for the careful consideration and high evaluation of the work.

To be relevant to the absorption of iodide ions it would be interesting to understand the selectivity. If the material is placed in a solution containing a mixture of anions which are absorbed preferentially? Is there selectivity in a complex mixture? 

Due to the presence of narrow and highly ordered channels in 3, its adsorption abilities are limited by the size of the anionic species. As we described in the manuscript, no inclusion or exchange of multinuclear anions, such as CN, NO2, I3, SeO42–, CrO42–, Cr2O72– were observed. Unfortunately, the investigation of the selective ion exchange between smaller mononuclear halide anions (Br, Cl, F) is not possible as the compound 3 decomposes in the solutions of the corresponding salts. Nonetheless, this predicament is less relevant in the context of the potential remediation of the nuclear wastes. Indeed, due to a very nature of the nuclear fission, the products of the radioactive decay of the heavy nucleus Z are known to have a distribution maximum at ca. Z/2 with a diminishing contribution of the heavier and lighter elements. That is why in the nuclear wastes of 92U or 94Pu the main competitors for the mononuclear 53I­ anion would be oxo/hydroxo complexes of f- and d-metals as well as oxoanions of the p-elements (antimonates, tellurates, iodates and xenates), all being multinuclear species and thus having quite a large size to fit the pores of 3. The concentration of the second most abundant mononuclear bromide anion (35Br) in a typical high-level nuclear waste is ca. one order of magnitude lower than for the iodide [Regulations for the Safe Transport of Radioactive Material, 2018 Edition, IAEA Safety Standards], therefore an I/Br ion selectivity is of minor issues, not to mention I/Cl and I/F selectivity having no practical relevance in the context of the selective extraction of iodide from the nuclear decay products.

More challenging would be to study the structural damage / integrity once radioisotopic iodide is absorbed.   

We agree to the reviewer that application of 3 as well as other MOFs in a practical adsorption of radioactive isotopes necessary requires extensive studies of stability of those materials upon the radiation, which are quite challenging by themselves and falls beyond the scope of our manuscript. Unfortunately, we do not have an access to the corresponding facilities to carry out such studies.

Concerning radiodegradation, most of the I radioisotopes undergo either β+ or β decay, which are known for the lowest particle energy (typically, less than 0.4 MeV) among other types of radiative decays and thus can bring a fatal damage only if absorbed by living organisms. Also, the compound 3 is constructed from relatively light atoms (27Co being the heaviest) and possesses rather sparse, spongy structure, which makes it poor medium for absorption of secondary γ-rays [10.1021/acs.iecr.9b06820; 10.1039/C6CC06878B; 10.1021/acs.langmuir.2c01074], generated by electron capture decay (occurring for 123I and 125I) or positron annihilation in a bulk solid. Based on the above considerations it can be assumed that the radiation stability of 3 as well as other porous MOFs is generally better than for other inorganic adsorbents, such as clays and zeolytes, due to porous structure with spatially separated metal ions being main radiation absorption centers.

Reviewer 3 Report

Four new metal-organic frameworks based on cobalt(II) salts and 10 1,4-diazabicyclo[2.2.2]octane N,N'-dioxide (odabco) within the frame of magnetic properties and anion exchange in odabco-based coordination networks were prepared and identified in this study. The author identified these samples using single-crystal X-ray crystallography and interpreted the magnetization of crystal structures 1, 3 and 4. Among them, ion exchange experiments revealed a significant uptake of iodide by crystal structures 3, which can have future applications in the absorption of radioactive iodide anions. This paper is quite interesting, and I believe it merits consideration for publication in Nanomaterials.

Regarding the typo in the abstract section, line 12, it should be corrected as follows: “Magnetization measurements were performed for 1, 3 and 4 and the obtained data were interpreted on the basis of their crystal structures.”

Author Response

Four new metal-organic frameworks based on cobalt(II) salts and 10 1,4-diazabicyclo[2.2.2]octane N,N'-dioxide (odabco) within the frame of magnetic properties and anion exchange in odabco-based coordination networks were prepared and identified in this study. The author identified these samples using single-crystal X-ray crystallography and interpreted the magnetization of crystal structures 1, 3 and 4. Among them, ion exchange experiments revealed a significant uptake of iodide by crystal structures 3, which can have future applications in the absorption of radioactive iodide anions. This paper is quite interesting, and I believe it merits consideration for publication in Nanomaterials.

Authors are grateful to the reviewer for high evaluation of the work.

Regarding the typo in the abstract section, line 12, it should be corrected as follows: “Magnetization measurements were performed for 1, 3 and 4 and the obtained data were interpreted on the basis of their crystal structures.”

The necessary correction has been made.

Reviewer 4 Report

The manuscript describes the synthesis and characterization of four novel MOFs. The authors conducted an extensive structural analysis of these compounds and demonstrated their ability to capture iodide. The topic is interesting, well-written, and the structural investigation is detailed. Therefore, I believe that the manuscript is suitable for publication.  Note, however, that porosity and surface area are known to be crucial parameters in MOF applications. Therefore, I recommend evaluating these parameters to enhance the paper's overall significance.

Author Response

The manuscript describes the synthesis and characterization of four novel MOFs. The authors conducted an extensive structural analysis of these compounds and demonstrated their ability to capture iodide. The topic is interesting, well-written, and the structural investigation is detailed. Therefore, I believe that the manuscript is suitable for publication.  Note, however, that porosity and surface area are known to be crucial parameters in MOF applications. Therefore, I recommend evaluating these parameters to enhance the paper's overall significance.

Authors are grateful to the reviewer for high evaluation of the work.

We fully agree to the reviewer that experimental porosity and surface area, typically derived from gas adsorption data, are important parameters for MOFs’ characterization. Carrying out such measurements requires a full deletion of guest moieties from the porous structure, typically achieved for neutral coordination frameworks under vacuum treatment. Cationic coordination networks in 3 and 4 possess notable specific void volumes, estimated by PLATON calculations as 26 % and 34 %, respectively, which are partially occupied by guest nitrate anions. These values can be recalculated to 0.16 cm3·g–1 (3) and 0.24 cm3·g–1 (4) porosities, according to the crystallographic densities of the compounds. Such porosity characteristics are suitable enough for the diverse applications in gas and vapor adsorption for neutral MOF adsorbents [10.1021/jacs.9b08322]. However, in case of charged coordination networks, their guest counterions cannot be deleted from the porous host due to the necessary electroneutrality of the overall structure. Subsequently, the porosity and surface area of narrow-pored ionic MOFs, such as 3 and 4, cannot be reliably measured experimentally because of the pore blockage arising from the strictly required occurence of counterions within their voids.

It should be noted that the presence of the guest nitrates in 3, which could apparently slam the possible gas adsorption characteristics, does not harm anyway its anion exchange properties. These were unambiguously confirmed in the present work by a significant uptake of I, driven by a substitution of the nitrates by iodides within an almost unperturbed porous host framework. Thus, besides the absence of gas adsorption data, the accessibility of voids in 3 for small anions is beyond doubt, providing its valuable applications as an anion adsorbent.
